# The Impact of PET/CT on Paediatric Oncology

**DOI:** 10.3390/diagnostics13020192

**Published:** 2023-01-05

**Authors:** Anita Brink, Khanyisile N. Hlongwa, Stuart More

**Affiliations:** Division of Nuclear Medicine, Department of Radiation Medicine, Faculty of Health Sciences, University of Cape Town, Cape Town 7700, South Africa

**Keywords:** paediatric oncology, PET/CT, lymphoma, neuroblastoma, sarcoma, Langerhans-cell Histiocytosis

## Abstract

This review paper will discuss the use of positron emission tomography/computed tomography (PET/CT) in paediatric oncology. Functional imaging with PET/CT has proven useful to guide treatment by accurately staging disease and limiting unnecessary treatments by determining the metabolic response to treatment. ^18^F-Fluorodeoxyglucose (2-[^18^F]FDG) PET/CT is routinely used in patients with lymphoma. We highlight specific considerations in the paediatric population with lymphoma. The strengths and weaknesses for PET/CT tracers that compliment *Meta*-[^123^I]iodobenzylguanidine ([^123^I]mIBG) for the imaging of neuroblastoma are summarized. 2-[^18^F]FDG PET/CT has increasingly been used in the staging and evaluation of disease response in sarcomas. The current recommendations for the use of PET/CT in sarcomas are given and potential future developments and highlighted. 2-[^18^F]FDG PET/CT in combination with conventional imaging is currently the standard for disease evaluation in children with Langerhans-cell Histiocytosis (LCH) and the non-LCH disease spectrum. The common pitfalls of 2-[^18^F]FDG PET/CT in this setting are discussed.

## 1. Introduction

The overall survival of children with cancer has improved significantly in the past decades, which has led to a shift in the management of childhood malignancies. Non-invasive imaging has assisted in the evaluation of children with a known or suspected disease. With the longer survival of these patients, it is clear that many patients suffer severe long term side effects from the treatments [1,2]. The treatment aim in paediatric oncology is to achieve a cure with the minimum long-term side effects. Functional imaging with positron emission tomography/computed tomography (PET/CT) has proven useful to guide treatment in paediatric malignancies by accurately staging disease, restaging, evaluating treatment response and limiting unnecessary treatments by determining the metabolic response to treatment and the prognostic significance thereof. In this paper, we review the most frequently imaged paediatric neoplasms seen in the clinical setting, e.g., brain tumours, osteosarcoma, Ewing Sarcoma, neuroblastoma and lymphoma. We look at the most commonly used PET tracer, ^18^F-Fluorodeoxyglucose (2-[^18^F]FDG) PET/CT and its utility in improving the management of the aforementioned neoplasms [3].

## 2. Paediatric Lymphoma

Lymphoma is one of the most prevalent paediatric malignancies [4]. Hodgkin lymphoma (HL) is common in adolescence. The most common histological HL subtype is nodular sclerosing HL. Non-Hodgkin lymphoma (NHL) is more common (60%) in children than HL (40%). Common NHL subtypes include Burkitt and Burkitt-like, precursor T-cell, diffuse large B cell, and anaplastic large cell lymphoma. Indolent lymphomas are rare in children [5]. 2-[^18^F]FDG PET/CT is now established as the reference investigation for paediatric patients with HL and NHL [6].

2-[^18^F]FDG PET/CT is routinely used for the staging of paediatric HL and NHL. Patients may have bulky disease causing airway compression. Care should be taken to assess for airway compromise prior to sedation. 2-[^18^F]FDG PET/CT has an excellent sensitivity for detecting nodal disease, including splenic involvement, as well as solid organ involvement, such as bone, liver, and lung [5].

Waldeyer’s ring and posterior cervical nodes can be difficult to interpret in children, especially on the interim and end of treatment 2-[^18^F]FDG PET/CT. Children are prone to viral and bacterial infections which frequently involve the upper air ways. Recent chemotherapy reduces the patient’s immunity further, which increases the risk for infections [5]. The use of 2-[^18^F]FDG PET/CT in combination with MRI is useful to identify pathology in Waldeyer’s ring [7].

Children frequently have increased thymic activity on baseline scans and an increased uptake in response to treatment, thymic rebound, is also commonly seen. It is important to differentiate between normal and pathological uptake [8]. Focal increased uptake in a gland which has abnormal contours or a nodular appearance on CT is suspicious for thymic disease [9]. Low-grade increased splenic and bone marrow uptake may also occur usually on the interim or end of treatment scans as a physiological response to treatment [10].

2-[^18^F]FDG PET/CT is well established as a method with high diagnostic accuracy to assess bone marrow involvement. A recent meta-analysis reviewed the use of 2-[^18^F]FDG PET/CT and bone marrow biopsy for the detection of bone marrow involvement in paediatric patients. Nine studies were included: four with only HL, one with only NHL, and four with a mix of HL and NHL patients. The combined patient numbers were 1640 patients, 326 of whom had bone marrow involvement. The pooled sensitivity and specificity of 2-[^18^F]FDG PET/CT was 0.97 (95% CI: 0.93–0.99) and 0.99 (95% CI: 0.98–0.99), respectively. The pooled sensitivity and specificity of bone marrow biopsy was 0.44 (95% CI: 0.34–0.55) and 1.00 (95% CI: 0.92–100) [4]. There is a shift away from using bone marrow biopsies to establish bone marrow involvement in clinical HL trials [5].

The use of interim 2-[^18^F]FDG PET/CT to assess disease response in adult patients is well established. The treatment response for HL is measured on the interim 2-[^18^F]FDG PET/CT done after two cycles of first line treatment. Uptake is assessed using the 5-point Deauville criteria a visual score comparing uptake in the lesion to liver uptake. The Deauville 5-point score is as follows: 1 = no uptake above background; 2 = uptake ≤ mediastinum; 3 = uptake > mediastinum but ≤ liver; 4 = uptake greater than liver; 5 = uptake markedly higher than liver (2–3 times SUVmax in normal liver) and/or new lesions; X = new areas of uptake unlikely to be related to lymphoma [11].Patients with a Deauville score ≤ 3 have a favourable response to treatment, Figure 1 [12]. There are fewer studies investigating the utility of interim 2-[^18^F]FDG PET/CT in children. A prospective study with 57 paediatric patients with HL showed that the specificity for predicting relapse on an interim 2-[^18^F]FDG PET/CT using the 5-point Deauville criteria was 91.4% [13]. In low risk and intermediate risk paediatric patients with a favourable response to treatment on interim 2-[^18^F]FDG PET/CT, it has been shown that involved field radiotherapy may be omitted with no negative impact on patient outcomes [14].

The utility of quantitative PET (qPET) in paediatric lymphoma 2-[^18^F]FDG response assessment has been studied in 898 patients after 2 cycles of chemotherapy. The qPET is the quotient of the SUVpeak of the hottest residual over the SUVmean of the liver. The qPET methodology adds semi-automatic quantification for interim FDG-PET response in lymphoma as an extension of the Deauville criteria. In this cohort, a qPET < 1.3 excluded abnormal response with high sensitivity on interim 2-[^18^F]FDG PET/CT [15]. This has been subsequently utilised in other studies after this initial validation [16,17,18].

The end of treatment 2-[^18^F]FDG PET/CT has a slightly higher specificity (95.7%) for predicting relapse than the interim 2-[^18^F]FDG PET/CT(91.4%). A positive end of treatment 2-[^18^F]FDG PET/CT scan is associated with a significantly poorer overall survival [5,13].

Immune-base therapies can lead to flare/pseudoprogression of lesions on follow-up PET/CTs. It is important to differentiate true from pseudoprogression. This has led to the development of the new LYRIC criteria as an indeterminate-response category [11].

There is growing evidence for the utility of using machine learning methods for semiquantification, such as convolutional neural networks (CNN). Recent work by Etchebehere et al. demonstrated clearly that the determination of whole body tumour burden in patients with paediatric lymphoma using CNN is fast and feasible in implementation in clinical practice [19].

## 3. Neuroblastoma

### 3.1. 2-[^18^F]FDG PET/CT in Neuroblastoma

*Meta*-[^123^I]iodobenzylguanidine ([^123^I]mIBG) is currently still the preferred imaging agent for the staging and follow-up of patients with neuroblastoma. Due to differing tumour biology, 10% of all patients with neuroblastoma do not take up [^123^I]mIBG [20]. Uptake in lesions may also be discordant due to heterogenous tumour biology. 2-[^18^F]FDG PET/CT is a valuable tool in children with [^123^I]mIBG-negative disease or where there are discordance in uptake between [^123^I]mIBG and anatomical images [21].

If a lesion is not [^123^I]mIBG-avid at diagnosis, it is associated with better prognosis due to better tumour biology and lower proliferation index. As the tumours are treated, it may also lose its [^123^I]mIBG avidity due to tumour differentiation, and turn into a more benign pathology such as ganglioneuroma, 30% of which do not take up [^123^I]mIBG [21]. In a second group of patients, it may be an indication of a worse prognosis due to tumour dedifferentiation, especially if there is discordance between the [^123^I]mIBG and 2-[^18^F]FDG uptake. Higher 2-[^18^F]FDG has been found to be associated with poorer prognostic markers, such a N-MYC amplification [22].

[^123^I]mIBG is not normally taken up in bone or bone marrow; therefore, if abnormal uptake is seen in skeletal elements, the findings are in keeping with metastatic disease involving the bone marrow with or without cortical bone involvement. This is one of the reasons why [^123^I]mIBG is considered superior to 2-[^18^F]FDG for the detection of bone marrow uptake, as 2-[^18^F]FDG has low grade diffuse uptake in normal bone marrow [23]. Despite this apparent weakness of 2-[^18^F]FDG for the detection of bone marrow involvement, pathological 2-[^18^F]FDG in bone marrow is a reliable predictor of bone marrow involvement. This holds true especially for the baseline imaging. Diffuse increased in bone marrow uptake on an interim of post treatment 2-[^18^F]FDG PET/CT is part of the treatment response and may be falsely interpreted as pathological uptake [24].

2-[^18^F]FDG is better than [^123^I]mIBG for the evaluation of soft tissue disease. This is partly due to the higher spatial resolution as well as the routine addition of the CT component, which helps to differentiate between physiological and pathophysiological uptake in soft tissues [23].

Another important limitation of 2-[^18^F]FDG is the high physiological uptake of 2-[^18^F]FDG in the brain, which hampers the evaluation of skeletal involvement in the skull, a frequent site of metastatic disease [21].

### 3.2. [^124^I]mIBG PET/CT in Neuroblastoma

[^124^I]mIBG a PET/CT tracer has not entered the realm of routine clinical use due to a number of limitations. This tracer has a relatively long half-life, 4.2 days, which means that is it still a two-day imaging procedure. It has a positron abundance of only 23%, which leads to a relatively low image quality compared to other PET/CT tracers. There is also still a need to block the thyroid gland prior to imaging [21].

### 3.3. [^68^Ga]DOTA-SSA PET/CT in Neuroblastoma

Somatostatin receptors, specifically somatostatin receptor subtype 2 (SSTR2), are overexpressed in paediatric tumours. [^68^Ga]DOTA-somatostatin analogues ([^68^Ga]DOTA-SSA) bind to somatostatin receptors and are used to image tumours with overexpression of these receptors [25].

In laboratory studies, it has been shown that 77 to 89% of neuroblastoma cells express somatostatin receptors. The use of [^68^Ga]DOTA-SSA PET/CT in neuroblastoma is promising. In a small study [^68^Ga]DOTA-SSA uptake was confirmed in eight children with relapsed neuroblastoma. Four of these children went on to receive peptide receptor radionuclide therapy (PRRT) [26]. The largest study available at this time is a study by Gains et al. They imaged 42 patients with neuroblastoma with both [^123^I]mIBG and [^68^Ga]DOTATATE; all patients had [^68^Ga]DOTATATE uptake and two did not have [^123^I]mIBG uptake. They did find some lesion variation, with some lesions taking up [^123^I]mIBG and not [^68^Ga]DOTATATE and vice versa [27].

The use of [^68^Ga]DOTA-SSA in neuroblastoma has a number of advantages compared to [^123^I]mIBG. The imaging protocol is easier, with imaging being done 1 h after injection compared to the 24 h images of [^123^I]mIBG. Patient preparation is also easier, as there is no need to block the thyroid [28,29]. There is a long list of medications that interfere with [^123^I]mIBG uptake, including medications commonly used in children, such as over the counter cold and flu preparations [30]. Somatostatin therapy, which is rarely used in children, is the only treatment that needs to be stopped prior to [^68^Ga]DOTA-SSA imaging [29]. ^68^Gallium is obtained from a generator and is therefore readily available in centres with the ^68^Germanium/^68^Gallium generator; in contrast, [^123^I]mIBG is cyclotron-produced, which hampers availability in centres far from a cyclotron. Finally the special resolution with PET tracers is better than that of [^123^I]mIBG, Figure 2 [21].

If a tumour is ([^68^Ga]DOTA-SSA-avid, this opens up the possibility of PRRT with either [^177^Lu]DOTATATE or [^90^Y]DOTANOC [31]. There is growing interest in the use of PRRT in these patients [32,33].

### 3.4. [^18^F]F-DOPA PET/CT in Neuroblastoma

L-dihydroxyphenylalanine (L-DOPA) is a precursor of dopamine and it can be labelled with ^18^F to form [^18^F]Fluoro-L-DOPA ([^18^F]F-DOPA). [^18^F]F-DOPA is actively transported into cells via the large neutral amino acid transporter 1(LAT1) system and converted in the cell to [^18^F]fluorodopamine by the enzyme L-amino acid decarboxylase (AADC). In neuroblastoma, uptake of [^18^F]F-DOPA in tumour cells is due to increased intracellular transport and increased activity of AACD [21].

In a head-to-head comparison between [^123^I]mIBG scintigraphy and [^18^F]F-DOPA PET/CT, 19 patients with 28 paired studies, [^18^F]F-DOPA PET/CT outperformed [^123^I]mIBG scintigraphy. In total, 16 patients had [^18^F]F-DOPA-avid tumours in comparison to 11 patients with [^123^I]mIBG-avid tumours. On a lesion-based analysis [18F]F-DOPA PET/CT detected 141 lesions in comparison to the 88 lesions detected by [^123^I]mIBG scintigraphy. The lesion-based sensitivity of [^18^F]F-DOPA PET/CT was 90% vs. 56% for [^123^I]mIBG scintigraphy. There was no difference in specificity [34]. Piccardo et al. compared [^18^F]F-DOPA PET/CT to [^123^I]mIBG SPECT/CT in a prospective study in children and found that [^18^F]F-DOPA PET/CT was more sensitive in detecting soft tissue metastases and small bone localizations; they also found it to be better for post treatment evaluation and for prognostic significance. [35] Comparing 2-[^18^F]FDG with [^18^F]F-DOPA, higher [^18^F]F-DOPA after induction treatment is associated with a better prognosis [36]. Despite the apparent advantages of [^18^F]F-DOPA PET/CT, its role is currently limited to the small number of patients who do not have [^123^I]mIBG uptake [21].

### 3.5. Other Tracers in Neuroblastoma

Meta-[^18^F]Fluorobenzylguanidine ([^18^F]mFBG]) has a similar uptake mechanism to [^123^I]mIBG [21]. This is still and experimental tracer and has been reported in the preclinical setting as well as in a small study which reported uptake three-times higher than in-vivo [^123^I]mIBG in neuroblastoma [37]. A small clinical study by Pandit-Taskar et al. conducted the first-in-human use of this tracer, of which five patients had neuroblastoma. This study demonstrated the benefits of single day imaging versus two days with [^123^I]mIBG, which will be beneficial in the paediatric setting, and improved spatial resolution and image detection. This will be of great benefit once the tracer becomes more readily available [38].

Carbon-11(^11^C)-labeled meta-hydroxyephedrine ([^11^C]mHED is a noradrenaline analogue that has been investigated for use in neuroblastoma. In limited studies this tracer has shown higher diagnostic accuracy than [^123^I]mIBG. The use of this tracer is limited by its short half-life and the need for an onsite cyclotron. The advantage of this tracer is lower radiation exposure due to its short half-life [21,39,40]. The different tracer for use in neuroblastoma is summarised in Table 1.

## 4. Sarcoma

### 4.1. Osteosarcoma

Osteosarcoma is the most prevalent paediatric bone tumour with an incidence of 4.8 per million per year [41]. It frequently metastasizes to the bones and lungs. 2-[^18^F]FDG PET/CT is more sensitive than [^99m^Tc]Tc-methylene diphosphonate ([^99m^Tc]Tc-MDP]) bone scintigraphy for the detection of bone metastases. In the largest study comparing 2-[^18^F]FDG PET/CT and [^99m^Tc]Tc-MDP in 206 patients, 38 of whom had bone metastases the sensitivity, specificity and diagnostic accuracy of [^18^F]FDG PET/CT was 95, 98 and 98%, and the corresponding values for [^99m^Tc]Tc-MDP scintigraphy were 76, 97 and 96%, respectively. One of the weaknesses of bone scintigraphy identified in this study were the detection of lesions close to the growth plates [42]. A number of smaller studies also confirmed superiority of 2-[^18^F]FDG PET/CT for the detection of bone metastases [43].

Dedicated lung CT scans are still the investigation of choice for the evaluation of lung metastases. 2-[^18^F]FDG PET/CT has a lower sensitivity for detecting small lung metastases [43,44]. 2-[^18^F]FDG avidity can be useful in differentiating between benign and malignant lesions. However, if a lung lesion is 2-[^18^F]FDG-avid, standard uptake value (SUV) > 1, it is more likely to be a lung metastasis [45].

Detecting disease recurrence at the site of surgical resection in the region of the limb prosthesis is one the most difficult areas in medical imaging. There is a foreign body reaction with low grade 2-[^18^F]FDG uptake around the limb prosthesis which may persist for a long period of time. Uptake around the prosthesis is usually more intense at 12 and 18 months after surgery than 3 months after surgery. If there is high 2-[^18^F]FDG uptake, SUVmax > 4.2, at the time of suspected recurrence and if uptake increases significantly with successive scans, ∆SUV > 75.0, 2-[^18^F]FDG PET/CT has a sensitivity, specificity and accuracy for detecting local recurrence of 78, 94 and 93% [46].

A good histopathologic response, >90% necrosis, after neoadjuvant chemotherapy, is a predictor of favourable response to treatment [46]. Using 2-[^18^F]FDG PET/CT as a surrogate investigation prior to surgery to assess the response to neoadjuvant treatment has been investigated. A significant decrease in 2-[^18^F]FDG uptake from baseline to the end of neoadjuvant chemotherapy is an independent predictor of disease and histological response. Patients with persistent high or increasing 2-[^18^F]FDG uptake despite neoadjuvant treatment have a poorer progression towards survival [47]. The use of 2-[^18^F]FDG to adapt treatment strategies in children with poor response to neoadjuvant chemotherapy still needs to be investigated further [44].

### 4.2. Ewing Sarcoma

Ewing sarcoma is the second most common paediatric bone tumour with an incidence of 3 per million per year [43]. It usually arises from bones but may also arise from soft tissue. Unlike osteosarcoma [48], it also metastasizes to soft tissue and bone marrow. The most frequent metastatic lesions occur in lung and bone [49].

Conventional staging for Ewing sarcoma still includes [^99m^Tc]Tc-MDP bone scintigraphy [44].

Early studies showed that 2-[^18^F]FDG PET is superior to [^99m^Tc]Tc-MDP bone scintigraphy for detecting bone metastases. In a study with 38 patients with Ewing sarcoma, the examination based sensitivity, specificity and diagnostic accuracy was 1.00, 0.96 and 0.97 for 2-[^18^F]FDG PET and 0.68, 0.87 and 0.82 for bone scintigraphy [50]. A prospective multicenter study, which included 46 paediatric patients with Ewing sarcoma, compared 2-[^18^F]FDG PET with conventional imaging, magnetic resonance imaging (MRI), CT and bone scintigraphy, and found that 2-[^18^F]FDG PET was superior for detecting lymph node (sensitivity 92% vs. 25%) and bone involvement (90% vs. 57%) [51].

Hybrid PET/CT imaging is superior to standalone PET imaging. A small study, 13 patients including six patients with metastatic bone disease, found that 2-[^18^F]FDG PET/CT detected more bone lesions but missed two lesions in the skull due to high physiological FDG uptake in the brain [52]. Another small study showed that 2-[^18^F]FDG PET/CT detected bone metastases in 11 of 12 patients with bone metastases; the false negative study was in a patient with a sclerotic bone lesion. In comparison, bone scintigraphy detected bone metastases in nine of the 12 patients; the three false negative studies occurred in patients with lytic bone lesions [53].

A meta-analysis which included 31 studies and 735 patients found that the pooled sensitivity of 2-[^18^F]FDG PET and 2-[^18^F]FDG PET/CT for the detection of bone metastases was 83.9% (95% CI: 70.5–91.9%) the pooled specificity was 93.2% (95%CI: 86.9–96.6%). The overall sensitivity for detecting lymph node metastases was 79.3% (95% CI: 58.7–91.2%) and the specificity was 97.9% (95% CI: 93.5–99.3%). Not surprisingly, the overall sensitivity for detecting lung metastases was low at 76.1% (95% CI: 61.4–86.5%), and the overall specificity was 92.4% (95% CI: 86.3–95.9%) [54].

Ultimately, 2-[^18^F]FDG PET/CT is able to reliably detect bone marrow involvement and some authors are recommending that routine bone marrow aspiration at diagnosis may be omitted if 2-[^18^F]FDG PET/CT is available [55,56,57].

2-[^18^F]FDG PET/CT is recommended for restaging relapsed patients. A meta-analysis of 5 trails involving 123 patients found that 2-[^18^F]FDG PET and 2-[^18^F]FDG PET/CT had a pooled sensitivity of 93% (95% CI: 83–98%) and a pooled specificity of 95% (95% CI: 80–96%) [58], Figure 3.

The value of 2-[^18^F]FDG PET/CT to predict treatment response and overall survival is still under investigation. If a patient has a positive 2-[^18^F]FDG PET/CT at relapse, it is associated with a shorter overall survival [59]. In a study of 28 patients with non-metastatic Ewing sarcoma the tumour SUV, metabolic tumour volume (MTV), and total lesion glycolysis (TLG) were measured at baseline and after completion of neoadjuvant chemotherapy. A ∆TGL with a cut-off of −60% was the best predictor of histological response, with a 100% sensitivity and a 77.8% specificity. However, SUV at the completion of neoadjuvant chemotherapy of >3.3 and a ∆TGL of less than −18% were independent predictors of worse overall survival on multivariate analysis [60].

Several other studies looking into the predictive role of 2-[^18^F]FDG PET/CT in Ewing sarcoma demonstrated heterogeneous results. A study by Raciborska et al., which looked at 50 patients with Ewing sarcoma that had imaging done at diagnosis and prior to induction of chemotherapy to assess histological response and PET metrics, demonstrated a positive correlation between SUV at diagnosis and response after neoadjuvant chemotherapy. Patients with SUV > 2.5 had a higher risk of relapse and death [61]. A relatively similar study also looked at 2-[^18^F]FDG PET/CT metabolic indices in assessment of histological response to neoadjuvant chemotherapy in 31 patients with Ewing sarcoma and osteosarcoma did not demonstrate a predictive role; however, it confirmed that 2-[^18^F]FDG PET/CT was superior in detecting skeletal and soft tissue lesions and that conventional imaging was superior in detecting small pulmonary metastases, as seen in Figure 3 [62]. A children’s hospital in Egypt also looked to assess the predictive value of 2-[^18^F]FDG PET/CT parameters in histological response to neoadjuvant chemotherapy and recommended a SUVmax cut off value of <2.5 to predict histological response. This, however, had a wide confidence interval. The limiting factors in these studies are the retrospective nature and small size of the studies, which make inferences on the predictive role of 2-[^18^F]FDG PET/CT metabolic parameters difficult to conclude with certainty in the absence of larger prospective studies [63].

Additional imaging of primary bone tumours utilizing PET/CT has been explored in adults with [^18^F]NaF (Sodium Flouride), which has similar characteristics to [^99m^Tc]Tc-MDP but improved resolution. The benefits are the half-life and imaging an hour after injection in comparison to three to four hours with [^99^mTc]Tc-MDP, and improved sensitivity. The role of [^18^F]NaF in primary bone tumours in paediatrics is not well-established [64,65].

### 4.3. Rhabdomyosarcoma

Rhabdomyosarcoma is the most common soft tissue sarcoma in children and adolescents. This tumour can metastasize to lymph nodes, lung, bone marrow and bone [44].

In 2021, a Cochrane review assessing the usefulness of 2-[^18^F]FDG PET/CT in staging of rhabdomyosarcoma based on only two studies with a total of 36 patients concluded that there was no convincing evidence for the use of this modality for the staging in this tumour and that larger studies were needed [66].

The European soft tissue sarcoma study group recently published a study comparing the use of 2-[^18^F]FDG PET/CT to the conventional imaging work-up at diagnosis in 118 children with metastatic rhabdomyosarcoma. 2-[^18^F]FDG PET/CT had a high sensitivity compared to conventional imaging work-up 96.2% vs. 78.5% for locoregional disease and 94.8% vs. 79.3% for distant lymph node involvement. 2-[^18^F]FDG PET/CT was more sensitive than bone scintigraphy (96.4% vs. 67.9%). The 2-[^18^F]FDG PET/CT sensitivity and specificity for detection of bone marrow involve was also high, at 91.8% and 93.8%, respectively. Dedicated CT lung was still the reference investigation for the detection of lung metastases. Importantly, in four patients, 2-[^18^F]FDG PET/CT upstaged the patients from localized to metastatic disease. One of the recommendations of this study was that 2-[^18^F]FDG PET/CT should replace bone scintigraphy for the evaluation of bone metastases [67].

### 4.4. Nephroblastoma

2-[^18^F]FDG PET/CT has not significant impact on the management of patients with nephroblastoma compared to other imaging modalities. Nephroblastoma frequently metastasizes to the lungs and PET/CT will unfortunately miss smaller lung metastases [68].

## 5. Pediatric Brain Tumours

The use of 2-[^18^F]FDG is limited in paediatric brain tumours due to the high physiological brain uptake of the tracer [69]. This tracer has proven to be useful in detecting distant metastases in patients with glioblastoma [70].

In the last couple of decades, other tracers have been developed which target different metabolic pathways. Malignant cells have high levels of choline-kinase activity with increased utilization of choline. A number of small studies have investigated the use of [^18^F]Fluoromethyl-choline ([^18^F]FCH) and ^18^F]Fluoroethyl-choline ([^18^F]FEC) PET/MRI in paediatric brain tumours [25].

Newer tracers are showing promise in the evaluations of paediatric brain tumours. These include tracers which bind to the gastrin releasing peptide receptor (GRPR), such as [^68^Ga]Ga-NOTA-Aca-BNN. ([^68^Ga]Ga-BNN) GRPR has increased expression in glioma. This also opens up the possibility of treatments with GRPR-targeted therapy [71].

A number of tracers have been investigated in this setting of paediatric brain tumours; these include, [^18^F]F-DOPA, [^18^F]Fluoro-ethyl-tyrosine ([^18^F]FET), [^11^C]methylmethiamine and [^64^Cu]CuCl2 [3,72,73].

Somatostatin receptors specifically somatostatin receptor subtype 2 (SSTR2) is overexpressed in paediatric tumours, including medulloblastoma and primitive neuroectodermal tumours (PNET). If a brain tumour is ([^68^Ga]DOTA-SSA-avid, this opens up the possibility of PRRT with either [^177^Lu]DOTATATE or [^90^Y]DOTANOC [31].

## 6. Langerhans-Cell Histiocytosis

Langerhans-Cell Histiocytosis (LCH) is one of the most common of the Histiocytosis. Globally, the incidence is 5–9 per million children under the age of 15 years [74]. LCH has a varied clinical presentation with patients presenting with solitary lesions to multisystemic disease [75]. The implications of imaging are important to characterize whether patients have low vs high-risk LCH, particularly with the association of BRAF V600E mutation [76].

The most common area affected In LCH is the skeleton (78–90%), with the skin and pituitary gland being affected in 33% and 25% of patients, respectively [77,78]. Other organs can be affected, such as the spleen, liver, haemopoetic system, lung (at 15%), lymph nodes (up to 10%) and central nervous system excluding the pituitary gland (2–4%) [74]. The use of 2-[^18^F]FDG PET/CT in LCH has grown significantly over the last decade in diagnosis, staging, and response assessment in patients with LCH, particularly in distinguishing single-system LCH (SS-LCH), low-risk multisystem LCH (MS-LCH), and multisystem LCH, which have different treatment implications for the referring clinician, as can be observed in Figure 4 [74,79].

2-[^18^F]FDG PET/CT is recommended in the current National Comprehensive Cancer Network (NCCN) guidelines for staging and response assessment [80]. Multiple studies have demonstrated the utility of 2-[^18^F]FDG PET/CT to detect more lesions than conventional imaging in multiple sites [81,82,83,84]. Jessop et al. reported a sensitivity and specificity of 100% and 83%, respectively, with low false positive rates [79]. The skull being one of the most affected skeletal sites; as such, detection of lesions in this area is challenging, with 2-[^18^F]FDG PET/CT owing to the physiological brain activity. Suspected CNS LCH involvement should ideally be characterized with MRI [85]. CT and MRI should continue to be used as adjunctive studies to characterize LCH lung involvement owing to discordant areas where there are positive lung findings for LCH with no 2-[^18^F]FDG uptake [81,82].

## 7. Other Disorders in the Histiocytoses Series

### 7.1. Rosai–Dorfman Disease

Rosai–Dorfman Disease(RDD) is a disease that predominantly affects children more than adults and typically presents with painless lymphadenopathy [75]. Patients are generally diagnosed on biopsy, with histology showing histiocytic cells positive for S100 and CD68 and negative for CD1a and CD207, unlike in LCH [86]. Extranodal manifestations of RDD is seen in approximately 40% of patients and can include the nasal cavity and paranasal sinuses (11%), skin which presents as subcutaneous masses (10%), bone (5–10%) and intrathoracic, retroperitoneal, genitourinary and CNS manifestations (<5%), as seen in Figure 5 [87].

2-[^18^F]FDG PET/CT is recommended as a baseline investigation in characterizing a disease [80]. In a single centre study where 109 studies were reviewed in 27 patients with RDD, distribution of disease was nodal/cutaneous in 18%, with predominance of extranodal involvement in the skeleton (33%), CNS (26%) and other extranodal sites (23%). 2-[^18^F]FDG PET/CT was also able to identify additional sites of disease in those patients who had prior anatomical imaging CT or MRI in 30% of patients. Of the 109 PET/CT studies, 13 led to a change in management affecting 41% of the patients. MRI is still recommended to characterize disease in the rare case of CNS, particularly if it involves the brain owing to high physiological 2-[^18^F]FDG uptake in this area [88].

### 7.2. Juvenile Xanthogranulomata

Of the Xanthogranulomata family, Juvenile (JXG) is the most common of the non-LCH spectrum, occurring predominantly in the first year of life [75,89]. Clinically, patients present with numerous red to yellow nodules up to 1cm in diameter which spontaneously resolve. Extracutaneous manifestations confer it a poor prognosis with a mortality rate of 5–10%; 2-[^18^F]FDG can assist stratifying extent of disease, despite there being no clear consensus recommendations on the preferred modality for monitoring therapy. Limited case reports have shown the utility of FDG to assist in staging and response assessment to therapy in multiple cutaneous and extracutaneous sites [90,91].

### 7.3. Haemophagocytic Lymphohistiocytosis

Haemophagocytic lymphohistiocytosis (HLH) is a rare disease characterised by intense immune activation owing to an excess of activated lymphocytes and macrophages, in which patients can present with fever, hepatomegaly, splenomegaly, hyperferritinaemia and cytopaenias. Owing to the severity of the clinical course, mortality is significant in these cases [75].

Primary HLH affects infants in an autosomal recessive mode. Secondary HLH(sHLH) can present at any age and is either associated with rheumatic disease, infection, malignancy, or prolongation of immunosuppression [75].

2-[^18^F]FDG PET/CT has been shown recently to assist in detecting potential malignancy and predicting prognosis in paediatric HLH with EBV infection. An SUVmax-lesions of >6.04, and SUVmax-Lymph node/Mediastinal blood pool > 5.74, with the presence of extranodal hypermetabolic lesions in multiple organs, was indicative of malignant HLH [92].

A second study also reviewed quantitative PET parameters to characterise malignant involvement in various organs in patients with sHLH. Malignant disease was considered when the SUVmax-lymph nodes was higher than 4.41 and other hypermetabolic lesions occurred in extranodal organs. In addition, with EBV-HLH, a higher SUVmax of the bone marrow was associated with a poorer prognosis [93].

## 8. Future Perspectives

The latest developments in PET/CT imaging is looking into a different aspect of the tumour microenvironment by imaging cancer associated fibroblasts (CAFs) with fibroblast activation protein (FAP) in comparison to tumour metabolism with 2-[^18^F]FDG PET/CT. FAP is a membrane-anchored serine protease with dipeptidyl peptidase and endopeptidase activity and is overexpressed by CAFs. CAFs are represented in many cancers and represent a diverse population of cells which promote or suppress tumour cells. Due to higher tumour uptake, low accumulation in normal tissues, and rapid clearance, FAPI PET/CT has demonstrated advantages over 2-[^18^F]FDG PET/CT; limitations such as physiological uptake by brain and liver in are not seen with FAPI PET/CT [94].

FAPI PET/CT has been demonstrated in up to 28 different kinds of cancer, including head and neck cancers, gastrointestinal tract cancers, pancreas and liver tumours, lymphoma, sarcoma, and gynaecological malignancies [95]. With relevance to tumours commonly seen in paediatric populations, FAPI PET/CT has demonstrated expression in bone and soft tissue sarcomas as well as in Hodgkin’s and Non-Hodgkin’s lymphomas. The imaging findings were compared to immunohistochemistry; however, the prognostic significance is yet to be explored [96,97].

To our knowledge, no studies have been published to date to demonstrate the utility of FAPI PET/CT in paediatric oncology; however, recent research shows that FAPI is a promising tracer, which will demonstrate value in imaging multiple malignancies as prospective studies comparing FAPI with conventional FDG, DOTA-SSA and DOPA are currently enrolled [98].

## Figures and Tables

**Figure 1 diagnostics-13-00192-f001:**
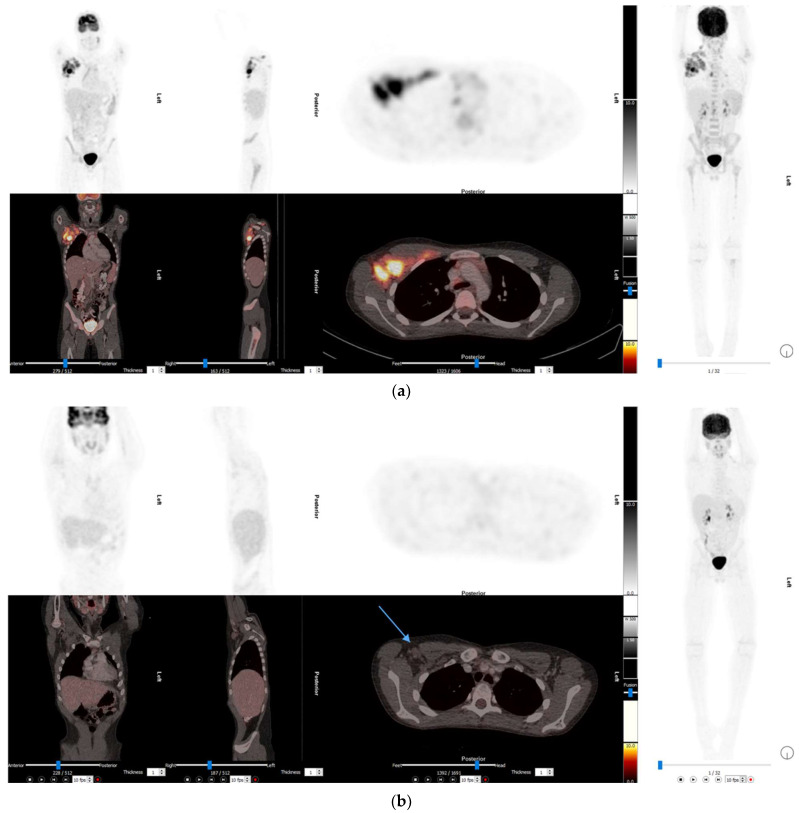
(**a**) The baseline 2-[^18^F]FDG PET/CT images of a 13-year-old boy with mixed cellularity HL, there is extensive abnormal uptake in a bulky right axillary lymph node mass, hilar and sub carinal nodes, subdiaphragmatic para-aortic nodes as well as extensive bone marrow involvement. Stage 4 disease. (**b**) The interim 2-[^18^F]FDG PET/CT performed after 2 cycles of induction chemotherapy. Nodes in the right axilla is smaller than before but, still enlarged. Uptake in these nodes is less than liver (Deauville score = 2); this is a complete metabolic response.

**Figure 2 diagnostics-13-00192-f002:**
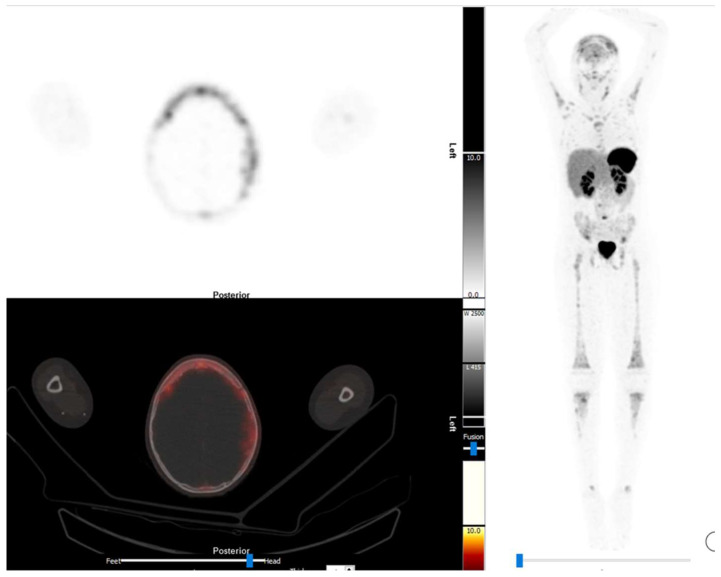
The [^68^Ga]Ga-DOTANOC images of a 11-year-old boy with stage 4 neuroblastoma which is refractory to treatment. There is widespread abnormal uptake in the skeleton. On the transaxial images illustrates extensive abnormal uptake in the expansile skull lesions.

**Figure 3 diagnostics-13-00192-f003:**
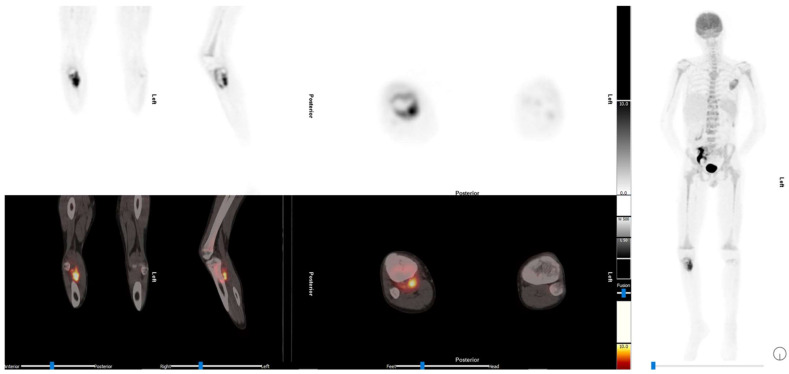
The 2-[^18^F]FDG PET/CT images of a 14-year-old with relapsed Ewing sarcoma. Diffuse increased bone marrow uptake in the long bones is in keeping with recent colony stimulating growth factor. He also had an autotransplant of a solitary kidney into the pelvis prior to radiotherapy. There are abnormal uptake bone lesions with accompanying soft tissue masses in the scapula and right proximal tibia. This illustrates the superiority of 2-[^18^F]FDG PET/CT for the detection of soft tissue masses.

**Figure 4 diagnostics-13-00192-f004:**
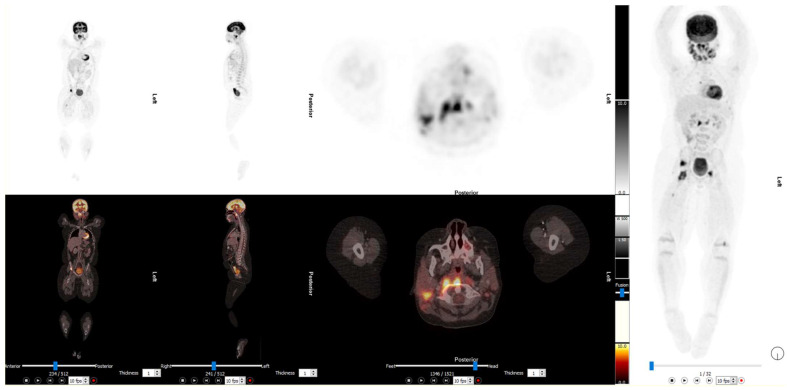
The 2-[^18^F]FDG PET/CT images of a 10-year-old with recurrent LCH being worked up for a stem cell transplant. There is evidence of significant disease in the nasopharynx with multiple skeletal lesions. This illustrates the value of 2-[^18^F]FDG PET/CT for re-staging prior to treatment.

**Figure 5 diagnostics-13-00192-f005:**
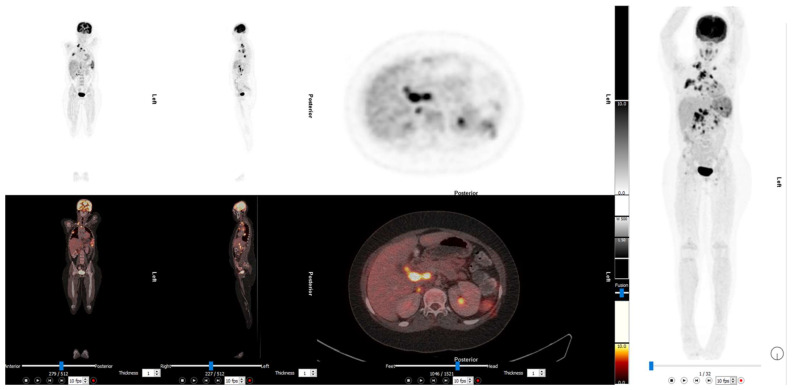
Restaging 2-[^18^F]FDG PET/CT images of a 12-year-old girl previously treated for Rosai–Dorfman disease. There is extensive abnormal lymph node uptake. Histology of one of the lesions revealed extra-pulmonary TB mimicking Rosai–Dorfman disease.

**Table 1 diagnostics-13-00192-t001:** Summary of PET tracers used for imaging neuroblastoma.

Radiopharmaceutical	Advantages	Disadvantages
2-[^18^F]FDG	Valuable in children with [^123^I]mIBG negative disease. Prognostic indicator Evaluates soft tissue disease Higher spatial resolution	Limited detection of bone marrow disease Physiological brain uptake hampers evaluation of skeletal involvement in the skull
[^124^I]mIBG	Long half-life Can be used for personal dosimetry	Low image quality Needs thyroid blockade Not widely available Higher radiation exposure
[^68^Ga]DOTA-SSA	Safe and feasible in children One day imaging due to short half-life Lower radiation Prognostic value Identify potential candidates for PRRT	Germanium generator required
[^18^F]F-DOPA	Prognostic value	Experience limited
[^18^F]mFBG]	Allows for single day, high resolution, quality imaging Shorter imaging advantages for children	Limited availability Lack of experience
[^11^C]mHED	Higher diagnostic accuracy Lower radiation exposure	Limited data Short half-life Limited availability Requires onsite cyclotron

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
