# Peer review of "The Impact of PET/CT on Paediatric Oncology"

_diagnostics, 2023, doi:10.3390/diagnostics13020192_

Round 1

Reviewer 1 Report

The authors comprehensively reviewed the clinical use of PET in pediatric malignancies, including lymphoma, neuroblastoma, sarcoma, and  Langerhans -cell Histiocytosis (LCH). The review is informative. However, some concerns are listed below:

1. Authors are encouraged to explain why they chose these four cancers. Are the authors selecting these four cancers according to the incidence? or other reasons?

2. Deauville 5-point scale is important in response evaluation of lymphoma. The authors may explain what each Deauville point means.

3. The authors described several PET tracers for neuroblastoma. To explain more clearly, please summarize and compare the pros and cons of these tracers in a table. Also, the authors only demonstrated the SSA PET in neuroblastoma; how about presenting other PET tracer images of neuroblastoma?

4. The authors compared the accuracy of FDG PET with MDP bone scan in pediatric sarcoma. How about fluoride PET? 

5. Langerhans-Cell Histiocytosis commonly affects skeletons. The authors may review the use of fluoride PET in this disease. Also, please add a figure to describe the use of FDG PET in Langerhans-Cell Histiocytosis.

Author Response

Comments from Reviewer 1

  • Comment 1: Authors are encouraged to explain why they chose these four cancers. Are the authors selecting these four cancers according to the incidence? or other reasons?

Response: Thank you for pointing this out. We chose these particular cancers as they are the most commonly seen indications for PET/CT imaging in paediatric nuclear medicine departments.

  • Comment 2:Deauville 5-point scale is important in response evaluation of lymphoma. The authors may explain what each Deauville point means

Response: We agree with this and have included the changes within the text.

  • Comment 3:The authors described several PET tracers for neuroblastoma. To explain more clearly, please summarize and compare the pros and cons of these tracers in a table. Also, the authors only demonstrated the SSA PET in neuroblastoma; how about presenting other PET tracer images of neuroblastoma?

Response: Thank you for pointing this out. We have added a table summarizing the PET tracer for neuroblastoma in the discussion. Unfortunately, our institution only has access to SSA PET for Neuroblastoma as it is a resource-limited setting.

  • Comment 4:The authors compared the accuracy of FDG PET with MDP bone scan in pediatric sarcoma. How about fluoride PET? 

Response: Thank you for this comment. Fluoride PET has been included in the manuscript.

  • Comment 5: Langerhans-Cell Histiocytosis commonly affects skeletons. The authors may review the use of fluoride PET in this disease. Also, please add a figure to describe the use of FDG PET in Langerhans-Cell Histiocytosis.

Response: Thank you for the insightful comment. We have a figure to describe the use of FDG PET in LCH. We described the use of Fluoride PET in sarcoma, however, no prospective data in using Flouride PET in LCH has been described.

Reviewer 2 Report

Topic is interesting:

1) Ln 8 and in Ln 10: definitions of PET/CT and 2-[18F]FDG are missing;

2) Ln 13: “is increasingly been” is not correct;

3) Introduction is not well developed, with lot of repetitions (words: patients and treatment); it must be enriched. For example you can explain what are the most common pediatric cancers, the crucial role of maging technologies in many phases of the disease and the the added value of FDG PET/CT over standard imaging in many phases of the disease;

4) For this topic review did you conduct a literature search? If yes, what strategy did you use? keywords used for your research? a flow chart of article selection process for this review? articles included and excluded?. A search strategy should be insert.

5) There are not any summary tables about articles cited, add it.

6) In [18F]F-DOPA PET/CT in Neuroblastoma you don't cite Piccardo et al. (2020) that recently conducted a the first prospective study in 18 neuroblastoma patients comparing [18F]F-DOPA PET-CT with [123I]mIBG imaging (scintigraphy and SPECT-CT) using histological results or anatomical imaging (CT or MRI) as the reference standard.

7) In “other tracers in Neuroblastoma”, about [18F]mFBG], you write “This is still and experimental tracer with no clinical experience in neuroblastoma”. Neeta Pandit-Taskar et al (2018) talked about ten patients (5 with neuroblastoma and 5 with paraganglioma/pheochromocytoma) that received 148–444 MBq (4–12mCi) of 18F-MFBG intravenously and concluded that “..Preliminary data show that 18F-MFBG imaging is safe and has favorable biodistribution and kinetics with good targeting of lesions. PET imaging with 18F-MFBG allows for same-day imaging of NETs. 18F-MFBG appears highly promising for imaging of patients with NETs, especially children with neuroblastoma”. Why didn’t you cite this article, for example?

8) For [11C]mHED you could spend some explanation more;

9) For pediatric brain tumor you dont’ cite some PET tracers such as 18F-DOPA, 18F-FET, 11C -MET and [64Cu]CuCl2.

Author Response

Comments from Reviewer 2

  • Comment 1: Ln 8 and in Ln 10: definitions of PET/CT and2-[18F]FDG are missing;

Response: Thank you for pointing this out, this has been corrected.

  • Comment 2: Ln 13: “is increasingly been” is not correct;

Response: Thank you for making us aware of this, this has been corrected.

  • Comment 3: Introduction is not well developed, with lot of repetitions (words: patients and treatment); it must be enriched. For example you can explain what are the most common pediatric cancers, the crucial role of imaging technologies in many phases of the disease and the added value of FDG PET/CT over standard imaging in many phases of the disease;

Response: Thank you for your insightful comments and recommendations. The introduction has been revised.

  • Comment 4: For this topic review did you conduct a literature search? If yes, what strategy did you use? keywords used for your research? a flow chart of article selection process for this review? articles included and excluded?. A search strategy should be insert.

Response: Thank you for this comment. We did not include this due to our article not being a meta-analysis of the latest data.

  • Comment 5: There are not any summary tables about articles cited, add it.

Response: Thank you for this comment. We did not include this due to our article not being a meta-analysis of the latest data.

  • Comment 6: In [18F]F-DOPA PET/CT in Neuroblastoma you don't cite Piccardo et al. (2020) that recently conducted a the first prospective study in 18 neuroblastoma patients comparing [18F]F-DOPA PET-CT with [123I]mIBG imaging (scintigraphy and SPECT-CT) using histological results or anatomical imaging (CT or MRI) as the reference standard.

Response: Thank you for pointing this out. We have included this reference.

  • Comment 7: In “other tracersin Neuroblastoma”, about [18F]mFBG], you write “This is still and experimental tracer with no clinical experience in neuroblastoma”. Neeta Pandit-Taskar et al (2018) talked about ten patients (5 with neuroblastoma and 5 with paraganglioma/pheochromocytoma) that received 148–444 MBq (4–12mCi) of 18F-MFBG intravenously and concluded that “..Preliminary data show that 18F-MFBG imaging is safe and has favorable biodistribution and kinetics with good targeting of lesions. PET imaging with 18F-MFBG allows for same-day imaging of NETs. 18F-MFBG appears highly promising for imaging of patients with NETs, especially children with neuroblastoma”. Why didn’t you cite this article, for example?

Response: Thank you for this insightful comment and for pointing this out. We have expanded on 18F-MFBG and cited the above article as an example.

  • Comment 8: For [11C]mHED you could spend some explanation more;

Response: Thank you for thus comment. We have expanded on this.

  • Comment 9: For pediatric brain tumor you dont’ cite some PET tracers such as 18F-DOPA, 18F-FET, 11C -MET and [64Cu]CuCl2.

Response:  Thank you for the recommendation. This has been added in the text.

Round 2

Reviewer 1 Report

My comments have been properly addressed.

Reviewer 2 Report

Your corrections and addiotions rased the level of your article.